# The spread of pESI-mediated extended-spectrum cephalosporin resistance in *Salmonella* serovars—Infantis, Senftenberg, and Alachua isolated from food animal sources in the United States

Cong Li[1]*, Heather Tate[1], Xinyang Huang[2,3], Chih-Hao Hsu[1], Lucas B. Harrison[1], Shaohua Zhao[1], Gamola Z. Fortenberry[4], Uday Dessai[4], Patrick F. McDermott[1], Errol A. Strain[5]

1 Center for Veterinary Medicine, U.S. Food and Drug Administration, Laurel, Maryland, United States of America, 2 Department of Nutrition and Food Science, University of Maryland, College Park, Maryland, United States of America, 3 Joint Institute for Food Safety and Applied Nutrition, Center for Food Safety Security Systems, University of Maryland,College Park, Maryland, United States of America, 4 Food Safety and Inspection Service, U.S. Department of Agriculture, Athens, Georgia, United States of America, 5 Center for Food Safety and Applied Nutrition, College Park, Maryland, United States of America

* cong.li@fda.hhs.gov

**Data Availability Statement:** The data underlying the results presented in the study are available

## Abstract

The goal of this study is to investigate the origin, prevalence, and evolution of the pESI megaplasmid in *Salmonella* isolated from animals, foods, and humans. We queried 510,097 *Salmonella* genomes under the National Center for Biotechnology Information (NCBI) Pathogen Detection (PD) database for the presence of potential sequences containing the pESI plasmid in animal, food, and environmental sources. The presence of the pESI megaplasmid was confirmed by using seven plasmid-specific markers (*rd*A, *pil*L, *Sog*S, *Trb*A, *ipf*, *ipr*2 and *Inc*FIB(pN55391)). The plasmid and chromosome phylogeny of these isolates was inferred from single nucleotide polymorphisms (SNPs). Our search resolved six *Salmonella* clusters carrying the pESI plasmid. Four were emergent *Salmonella* Infantis clusters, and one each belonged to serovar Senftenberg and Alachua. The Infantis cluster with a pESI plasmid carrying $bla_{CTX-M-65}$ gene was the biggest of the four emergent Infantis clusters, with over 10,000 isolates. This cluster was first detected in South America and has since spread widely in United States. Over time the composition of pESI in United States has changed with the average number of resistance genes showing a decrease from 9 in 2014 to 5 in 2022, resulting from changes in gene content in two integrons present in the plasmid. A recent and emerging cluster of Senftenberg, which carries the $bla_{CTX-M-65}$ gene and is primarily associated with turkey sources, was the second largest in the United States. SNP analysis showed that this cluster likely originated in North Carolina with the recent acquisition of the pESI plasmid. A single Alachua isolate from turkey was also found to carry the pESI plasmid containing $bla_{CTX-M-65}$ gene. The study of the pESI plasmid, its evolution and mechanism of spread can help us in developing

from NCBI Pathogen detection DB. https://www.ncbi.nlm.nih.gov/pathogens.

**Funding:** The authors have declared that no competing interests exist.

**Competing interests:** NO authors have competing interests.

appropriate strategies for the prevention and further spread of this multi-drug resistant plasmid in *Salmonella* in poultry and humans.

## Introduction

Non-typhoidal *Salmonella* is a leading cause of foodborne illness in the United States [1]. Salmonellosis is usually self-limited without requiring medical treatment. However, cephalosporins and ciprofloxacin are generally recommended to treat people with severe infections or people with increased risk of invasive infection [2]. The rise of multidrug resistance (MDR) in *Salmonella* represents a special challenge to choosing antimicrobial treatment options and their efficacy.

Antimicrobial resistance (AMR) genes are often located on plasmids and other mobile genetic elements that serve as vehicles for the spread of AMR genes between bacterial strains and genera [3]. In 2014, an extended spectrum cephalosporin-susceptible, emergent *Salmonella* Infantis (ESI) was reported in Israel. It carried a chromosomal mutation on *gyr*A gene conferring resistance to quinolones and a MDR megaplasmid (~280 kb), which was designated as plasmid of emerging-specific Infantis (pESI) [4]. This plasmid carried genes conferring resistance to tetracycline, sulfonamides, and trimethoprim. In 2015, Franco et al. reported that an extended-spectrum beta-lactamase (ESBL) positive, MDR *Salmonella* Infantis clone with a similar plasmid named as "pESI-like" was found in the Italian broiler chicken industry [5]. Compared to the original pESI, the pESI-like plasmid carried an ESBL gene $bla_{CTX-M-1}$, which leads to resistance to cephalosporin. This Infantis clone subsequently led to human infections in 2013 and 2014 [5].

After the first two reports, there have been multiple reports of related MDR-ESBL[+] *Salmonella* Infantis worldwide [6–9]. Among them, several variants of pESI-like plasmids carried by MDR-ESBL[+] *Salmonella* Infantis were reported in different regions around world: a European variant (particularly Italy), harboring the $bla_{CTX-M-1}$ gene [5, 10]; a Russian variant harboring the $bla_{CTX-M-14}$ gene [11]; and an American variant harboring the $bla_{CTX-M-65}$ gene [6, 12–14]. In addition to the antimicrobial resistance genes, all pESI and pESI-like plasmids carried virulence genes encoding for fimbriae clusters, yersiniabactin siderophore, toxin/antitoxin systems, and resistance genes for mercury and disinfectant [15]. *In vitro* and *in vivo* studies showed that the Infantis strains with the pESI-like plasmid were superior in biofilm formation, adhesion, and invasion into avian and mammalian host cells [4, 16]. The carriage of this plasmid by Infantis likely leads to the rapid spread of the recipient clones. The spread of these pESI-bearing Infantis strains is of particular concern because this plasmid can confer resistance to multiple antimicrobials, heavy metals, and antiseptics. Additionally, some *Salmonella* Infantis strains with pESI-like plasmid have been found carrying a colistin-resistance gene (*mcr*-1) [17, 18], albeit on a IncX4 plasmid, further limiting treatment options, since colistin is considered as a last resort for infection.

Our previous work and other studies have shown that once the pESI is acquired by a *Salmonella* Infantis strain it might spread quickly through clonal expansion because the plasmid provides strains with advantages of enhanced colonization and virulence. The American variant of the pESI-like megaplasmid is predominantly associated with one large cluster of *Salmonella* Infantis defined as an emergent *Salmonella* Infantis (ESI) clone [13, 20]. The ESI clones with pESI-like plasmid have become dominant among *Salmonella* isolated from poultry sources since they were first reported in the United States in 2014 from U.S. retail chicken and in 2019

from turkey. The ESI clones with pESI-like plasmid comprised 29% and 7% of all *Salmonella* isolated from U.S. retail chicken and turkey, respectively [13].

Although the pESI-like plasmids spread mainly through clonal expansion, experimental evidence has shown the plasmid has a potential to transfer horizontally [2, 21]. Recent findings point to the possible transfer of the pESI-like plasmid from *Salmonella* Infantis to *Salmonella* Senftenberg (https://www.fda.gov/animal-veterinary/national-antimicrobial-resistance-monitoring-system/narms-interim-data-updates) and *Salmonella* Muenchen, Agona, and, Schwarzengrund [22, 23]. Furthermore, great variability in AMR gene composition suggests that the plasmid may be changing structurally as it spreads [13, 24].

There have been no clear definitions for pESI and pESI-like plasmids. The pESI and pESI-like plasmid are highly homologous to each other, as previous publications reported [4, 5, 12, 13, 15, 19, 21]. Therefore, in this study, the term "pESI" is used to refer to all "pESI" or "pESI-like" plasmids.

The aim of this work was to utilize the Pathogen Detection (PD) browser to investigate over 100,000 environmental isolates among the half million *Salmonella* isolates in NCBI's PD database and look for the isolates which likely carry pESI plasmids and investigate the genetic changes of pESI-like plasmid during the clonal expansion and horizontal transfer events. The PD browser is a public web portal developed by NCBI. It collects Sequence Reads Archive (SRA) data and metadata from many surveillance programs such as National Antimicrobial Resistance Monitoring System (NARMS) as well as research projects, including those with WGS on environmental and clinical pathogen isolates. The embedded PD pipeline assembles the raw reads and clusters them with closely related strains based on whole-genome multi-locus sequence typing and Single Nucleotide Polymorphisms (SNPs). The clusters under the PD browser are defined by isolates that are within 50 SNPs of at least one other isolate in the cluster (https://www.ncbi.nlm.nih.gov/pathogens/pathogens_help/#data-processing [25]). The pipeline also screens the assembled genomes using AMRFinderPlus [26] to annotate known antimicrobial, anti-septic/ biocides, and anti-heavy metal resistance genes. We explored the possibility of screening AMR annotation for clusters which likely host pESI-like plasmids and document its presence in different environmental sources and animal hosts with available metadata in the PD database. We also investigated the genetic relatedness of American ESI strains and characterized the evolution and spread of the American pESI plasmid variant in food and animal sources sampled in the United States.

## Results

### The clusters with pESI plasmid

The initial query of the PD browser resulted in 10 SNP clusters. Eight of the 10 clusters include at least 3 isolates in the cluster (Table 1). Table 1 shows that among the eight clusters, six clusters carried pESI plasmid, including four *S.* Infantis clusters, one *S.* Senftenberg and one *S.* Alachua cluster. Almost all strains in the four Infantis clusters carried the pESI plasmid, ranging from 99.2% in cluster ESI-CTX-M-65, to 100% in the three other clusters. Forty-three of 48 (89.6%) of strains in *Salmonella* Senftenberg ESS-CTX-M-65 cluster carried the pESI plasmid. Only one (0.02%) in *S.* Alachua cluster PDS000050501.9 carried the pESI plasmid.

Table 1 shows that five of the six clusters with pESI carried a unique $bla_{CTX-M}$ allele. For example, PDS000028342.9 and PDS00113177.3 carried $bla_{CTX-M-14}$ only (named as ESI-CTX-M-14-1 and ESI-CTX-M-14-2), PDS000032463.103 (named as ESI-CTX-M-1) carried $bla_{CTX-M-1}$ only. PDS000089910.296 (named as ESI-CTX-M-65) is an exception. This cluster carried two different alleles, $bla_{CTX-M-15}$ and $bla_{CTX-M-65}$. Only 13 out of 5765 (0.2%)

**Table 1. The SNP clusters resulted from the search on Jan. 23$^{rd}$, 2023.** Details of clusters with pESI (In bold) are listed in S1 and S2 Tables.

| NO | Serovar | SNP cluster in Pathogen Detection Browser | Cluster Name in this study | Matched isolates | Matched clinical isolates | Matched environmental isolates | Total isolates | percentage of strains with pESI | bla$_{CTX-M}$ family gene | Location |
|---|---|---|---|---|---|---|---|---|---|---|
| **1** | **Infantis** | **PDS000089910.296** | **ESI-CTX-M-65** | **2996** | **1199** | **1634** | **11463** | **99.20%** | **bla$_{CTX-M-65}$, bla$_{CTX-M-15}$** | **Mainly Americas** |
| **2** | **Senftenberg** | **PDS000045773.44** | **ESS-CTX-M-65** | **21** | **4** | **17** | **48** | **89.50%** | **bla$_{CTX-M-65}$** | **US only** |
| 3 | Kentucky | PDS000027970.401 | N/A | 137 | 42 | 13 | 1858 | 0% | bla$_{CTX-M-55}$, blaCTX-M-14, bla$_{CTX-M-15}$, bla$_{CTX-M-104}$ | Global |
| **4** | **Infantis** | **PDS000032463.103** | **ESI-CTX-M-1** | **122** | **88** | **3** | **487** | **100%** | **bla$_{CTX-M-1}$** | **Mainly Italy, UK** |
| **5** | **Infantis** | **PDS000028342.9** | **ESI-CTX-M-14-1** | **6** | **1** | **1** | **16** | **100%** | **bla$_{CTX-M-14}$** | **UK, Cyprus** |
| **6** | **Infantis** | **PDS000113177.3** | **ESI-CTX-M-14-2** | **8** | **6** | **2** | **8** | **100%** | **bla$_{CTX-M-14}$** | **Russia** |
| 7 | Typhimurium | PDS000115660.19 | N/A | 3 | 2 | 1 | 622 | 0% | **bla$_{CTX-M-55}$, bla$_{CTX-M-65}$, bla$_{CTX-M-15}$, bla$_{CTX-M-130}$** | Global |
| **8** | **Alachua** | **PDS000050501.9** | **N/A** | **1** | **0** | **1** | **42** | **2.4%** | **bla$_{CTX-M-65}$** | **US only** |

bla$_{CTX-M}$ alleles are bla$_{CTX-M-15}$, 5687 (98.6%) are bla$_{CTX-M-65}$, the remaining 65 are unnamed bla$_{CTX-M}$ alleles (S1 Table).

These six pESI clusters are associated with different geographic regions. For example, ESI-CTX-M-65 strains were mostly found in the Americas (Table 1 and S1 Table) and only 163 out of 11463 (1.4%) found in other regions. Isolates from other Infantis clusters (ESI-CTX-M-1, ESI-CTX-M-14-1, ESI-CTX-M-14-2) were mainly found in west European countries, Cyprus, and Russia (Table 1).

Fig 1 shows that the phylogenies of the four *Salmonella* Infantis clusters with pESI plasmids were polymorphic, indicating they do not have a recent common ancestor and thus the pESI plasmid was transmitted horizontally. Although both ESI-CTX-M-14-1 and ESI-CTX-M-14-2 clusters carried the same ESBL gene, they did not share the same recent ancestors. The phylogenetic analysis revealed isolates from the same region were more likely to be related, as ESI-CTX-M-1 and ESI-CTX-M-14-1, are closer to each other than to other clusters from the Americas or Russia.

Fig 1 also shows that the pESI plasmid spreads mainly through clonal expansion, as there is a broad congruence of genetic relatedness between chromosomes and plasmids. The plasmid evolution mirrors the chromosome evolution pattern of the host bacteria. If the evolution of the plasmids was not affected by the host, it would not form clades that are similar in structure to the chromosomal tree of the plasmid carriages. Fig 1A shows that the groupings inside of clusters ESI-CTX-M-1, ESI-CTX-M-14-1 and ESI-CTX-M-14-2 are identical between trees referred from SNPs of chromosomes or plasmids. Fig 1 also shows that the horizontal transfer is not a frequent event since only one clade instead of multiple clades on the chromosomal or plasmid tree carried pESI with bla$_{CTX-M-65}$ circulated in the Americas. Fig 1A shows that the pESI plasmid evolves in a stepwise fashion, compared to the linear evolution of chromosomes in these clusters.

ESI-CTX-M-65 carried two different alleles of the bla$_{CTX-M}$ gene family, bla$_{CTX-M-15}$ in thirteen isolates from Peru and bla$_{CTX-M-65}$ in isolates from different parts of the Americas,

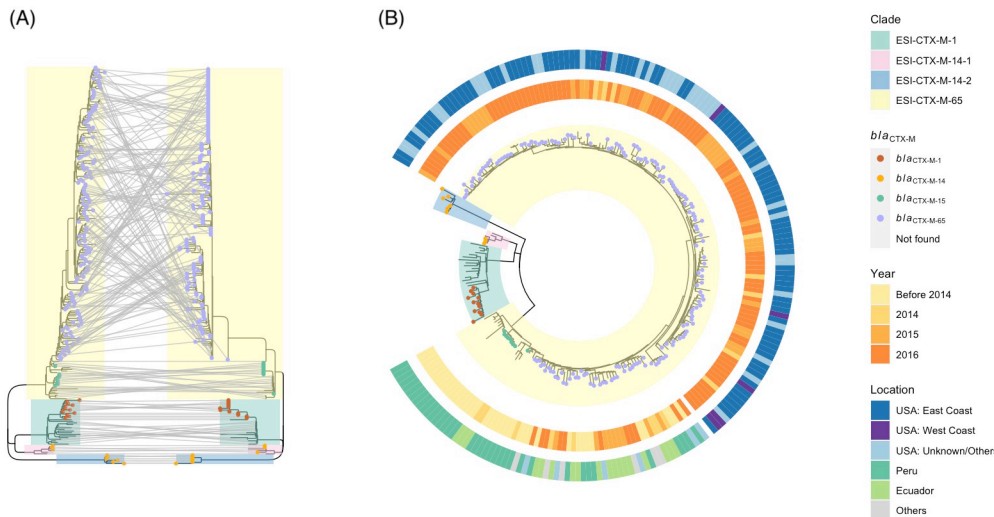

**Fig 1. The phylogeny of *Salmonella* Infantis strains (collected before 2016) with pESI plasmid from four clusters.**
(A) Comparison of chromosome (left) and plasmid (right) reference-based trees with Infantis strains collected before 2016 from four clusters in Table 1. The lines between two trees connect the same isolates. (B) Chromosomal reference-based tree with detailed meta data. The inner ring details the collection year; the outer ring details the location; USA: West Coast includes California, Oregon, Washington; USA:East Coast includes Delaware, Florida, Georgia, Maine, Massachusetts, New Jersey, New York, North Carolina, Pennsylvania, Virginia.

primarily the United States (Fig 1A). The strains with $bla_{\text{CTX-M-15}}$ appeared only in the earlier clades. To be noted, all thirteen isolates carried $bla_{\text{CTX-M-15}}$ gene (Fig 1B and S1 Table) are from South America, including nine clinical isolates from Peru (S1 Table). All isolates with $bla_{\text{CTX-M-65}}$ from United States appeared in newer clades, following the clades with Peruvian isolates carrying $bla_{\text{CTX-M-65}}$.

The chromosomal referenced-based phylogeny (Fig 1B) indicates that the ESI-CTX-M-65 may have originated in South America, as the isolates from early clades were all from Peru and Ecuador. Although the early food animal strains from 2014 and 2015 were primarily detected in eastern coastal states (46 from Maine, North Carolina, New Jersey, Virginia, and Maryland, 13 from Arkansas, Texas, and California), Fig 1B shows that isolates collected from west coast appeared in both earlier and later sub-lineages of ESI-CTX-M-65, even though they were collected in more recent years. This finding suggests that ESI-CTX-M-65 was introduced into the United States at various time at different locations, instead of it being spread to other states from the eastern coast.

## ESI-CTX-M-65 cluster

Among the six of *Salmonella* clusters carrying the pESI plasmid, the *S.* Infantis ESI-CTX-M-65 cluster is the largest cluster, with 11,463 isolates as of 1/23/2023. Among 11,203 isolates with source type information, 29.6% (3308) were clinical isolates and 70.5% (7,895) were environmental isolates. Approximately 95.4% (10,683/11203) of the isolates were collected from the Americas, with the majority (90.4%, 9,654 out of 10,683) collected in the United States. 212 of the 10,550 isolates from the Americas in this cluster with an available collection date were collected from 2009 to 2016 in Peru. The first poultry ESI-CTX-M-65 strain (SAMN09745937) from the United States was collected in December of 2013. This is earlier than previously

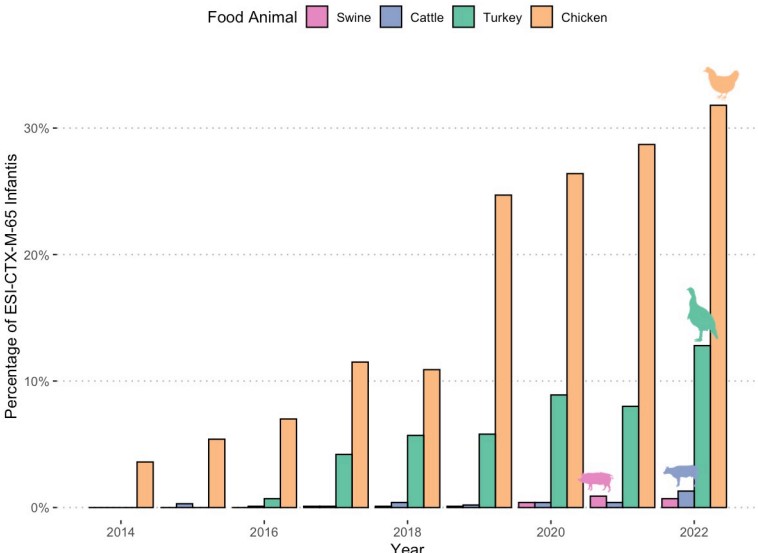

**Fig 2. The percentage of ESI-CTX-M-65 among *Salmonella* isolates from food animal sources collected in US.**

described [20]. The cluster currently (as of January 23rd of 2023) contains strains isolated from all 50 states.

Isolates in the ESI-CTX-M-65 cluster were checked for the presence of the pESI plasmids using the markers described in the Methods section. Of 8,290 genomes in ESI cluster, 7,640 (92%) had all seven markers; 425 (5%) had six markers; 136 (1.6%) had five markers; 49 (0.6%) had 4 markers; 39 had 3 markers; one had 2 markers. With one exception, all genomes in the ESI-M-65 cluster had the *Inc*FIB(pN55391) replicon. The one isolate without the *Inc*FIB replicon carried the six other markers (S1 Table). Using the presence of greater or equal to three markers as the criteria of pESI positive, 99.2% of these isolates carried pESI plasmid. Interestingly, the 40 isolates with 3 or fewer markers usually lose markers on the sequence originated from the *Inc*I plasmid. Notably, all 40 isolates were collected from 2017 or later, at least three years later than the first isolate with pESI appeared in the United States.

Using 41,280 strains in total from NARMS retail meat collection (BioProject 292661) and FSIS slaughter sampling (BioProjects 242847 and 292666), we calculated the proportion of strains from ESI-CTX-M-65 among all *Salmonella* strains from livestock sources and retail meat. We observed a temporal increase ranging between 0% and 31.8% in ESI-CTX-M-65 isolates among chicken, turkey, swine, and cattle between 2014 and 2022 (Fig 2), with significant increase among chicken isolates (from 3.6% to 31.8%) followed by turkey (0 to 12.8%) and cattle (0 to 1.3%). Notably turkey isolates did not appear until 2016 (Fig 2).

## The changes in the AMR gene content of pESI over time in ESI-CTX-M-65 cluster

Among the environmental and clinical isolates from South and Central America, there was an initial increase from an average of 5 plasmid-associated AMR genes in 2010 and 2011 to 9 in 2013 followed by a plateau (7.7 to 8.9 AMR genes) between 2013 and 2021 (Fig 3). This contrasted with the trend in NARMS human, food animal and retail meat isolates from the United States, where the average number of plasmid-associated AMR genes declined from 8.9–9.2 in 2014/2015 to 5.0 in 2022.

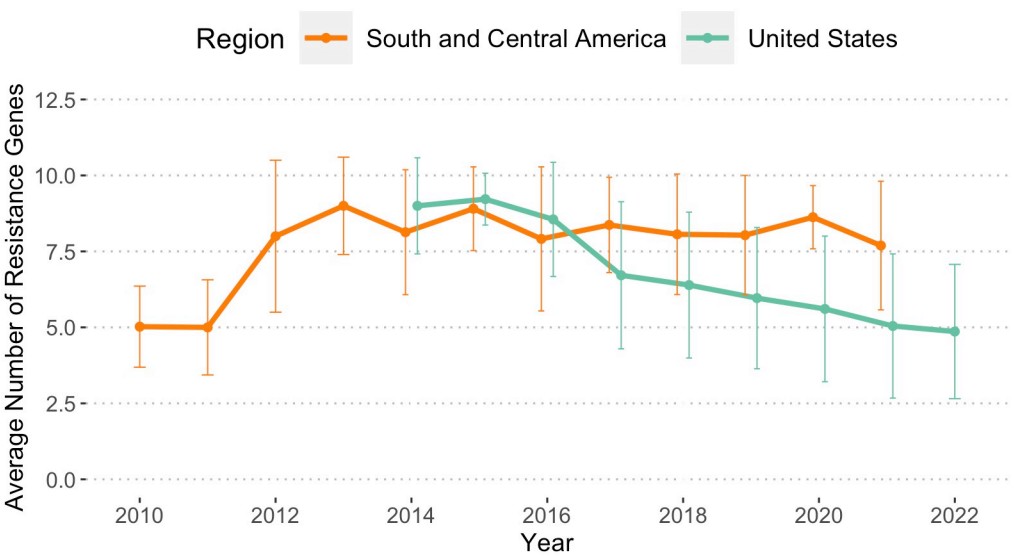

**Fig 3. The average number of AMR genes in ESI-CTX-M-65 strains in Americas.** The bars on each data point represent standard deviation.

We identified that AMR genes on the pESI plasmid located in two distinct regions organized by integrons. The Class I integron insertion site (Region 1): *aadA1-qacEdelta1-sul1--merE-merD-merC-merP-merT-merR-tet*(A); The Class 2 integron insertion site (Region 2): *aph*(3')-Ia-*dfr*A14-*aac*(3)-IVa-*aph*(4)-Ia-*flo*R-*fos*A3-$bla_{CTX-M-65}$ (Fig 4A).

To better understand the decline in the number of AMR genes in isolates from the United States, we looked at the AMR gene loci within each conserved integron insertion site. We found the rate of reduction across loci in the Region 1 (Fig 4A) were mostly unchanged, with the frequency of each gene consistently above 0.8 (Fig 4A). However, AMR genes loci were gradually lost in Region 2 as the plasmid expanded over the years (Fig 4A). Though there was a downward trend for all AMR genes within Region 2, the rate of reduction for individual AMR genes/loci was variable, with the most decline in *fos*A3 (from 0.76 in 2015 to 0.12 in 2022), followed by *aph*(3')-Ia, *dfr*A14, $bla_{CTX-M-65}$, *flo*R, *aac*(3)-IV and *aph*(4)-Ia (Fig 4A).

Fig 4B shows that while prevalence of resistance genes in Region 2 declined in all isolates between 2015 and 2022, their respective prevalence remained low in food animal and retail meat isolates compared to those in humans. For instance, the prevalence of $bla_{CTX-M-65}$ which was similar in both the groups in 2016, showed the biggest difference in 2022, with over 20% higher prevalence in human isolates (52.4%) compared to food animal sources (30.0%).

## Establishment of the pESI plasmid in *S.* Senftenberg

Our search of *Salmonella* clusters with pESI plasmid revealed a recently emerged *Salmonella* Senftenberg cluster (PDS000045773 named as ESS-CTX-M-65 in Table 1), with all isolates from 2019 or later (Fig 5).

Fig 5 shows that the three *Salmonella* Senftenberg isolates from the ESS-CTX-65 cluster that did not carry pESI plasmids belonged to an earlier lineage compared to other isolates with the pESI plasmid. Among the 27 isolates in this cluster that carried pESI, 15 carried an *Inc*Q plasmid containing four additional AMR genes (*sul*2, *aph*(3")-Ib, *aph*(6)-Id, *tet*(A)) (S1 Fig). Isolates in the Senftenberg cluster carried between 3 to 12 AMR genes (S2 Table). There were 23 isolates from food animals carrying pESI plasmid, all from turkey. Among them, 21 were

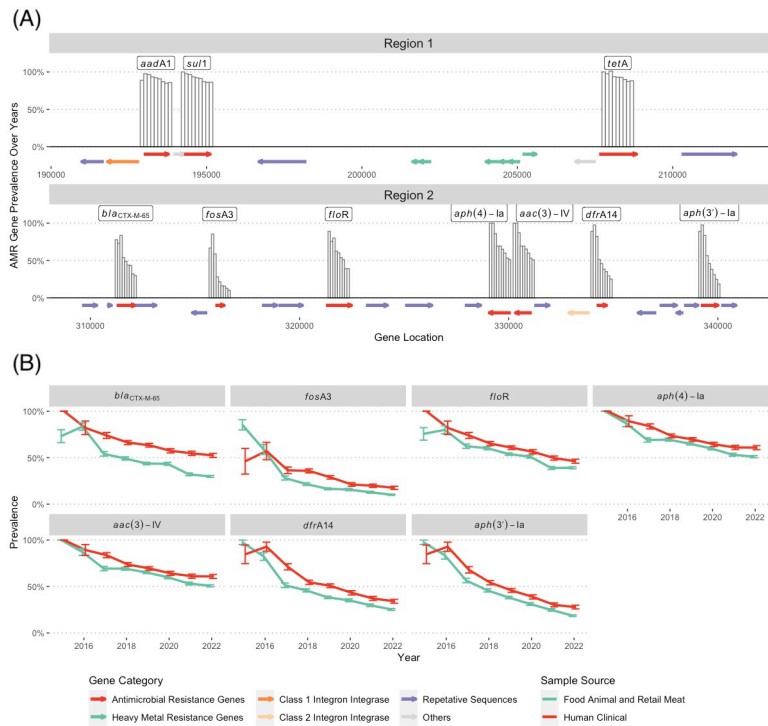

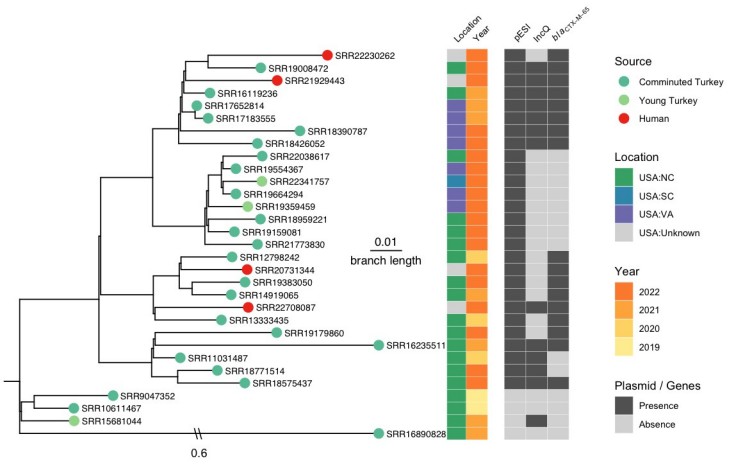

**Fig 4. The change of AMR gene in food animal sources and human strains between 2014–2022.** (A) the structure of the Region 1 and 2, and Prevalence of AMR genes in Region 1 and 2 between 2014 and 2022 from left to right; the bars represent the percentage of the isolates in a given year carried the gene indicated on the map. (B) Prevalence of AMR genes in Region 2. The isolates from 2012 to 2014 were excluded since there were less than ten isolates for each group to compare with. The data points represent the percentage of the clinal isolates (in red) and food animal and retail meat isolates (in green) in a given year carried indicated genes in each plot. The error bar was calculated by sqrt(prop* (1-prop)/n) where prop is frequency of the gene's presence. All the resistance gene presence information were obtained from the result of AMR Finder under Pathogen detection database (https://www.ncbi.nlm.nih.gov/pathogens/).

**Fig 5. The establishment of pESI in an emerging *Salmonella* Senftenberg cluster ESS-CTX-M-65.** The phylogenetic analysis includes all food animal isolates (26) and human (4) isolates in ESS-CTX-M-65. The tree was rooted with a turkey *Salmonella* Senftenberg isolate (FSIS22106145).

from comminuted turkey and 2 from young turkey. Thirteen of 23 (56.5%) turkey isolates carried a $bla_{\text{CTX-M-65}}$ gene, while all 4 (100%) human isolates carried pESI plasmid that had the $bla_{\text{CTX-M-65}}$ gene.

## Horizontal transferring events of pESI from *Salmonella* serovar Infantis to serovars Senftenberg and Alachua

The initial search for the *Salmonella* strains with pESI also resulted in a single *Salmonella* Alachua isolate FSIS22029592 with pESI plasmid from the cluster PDS000050501.9 (Table 1). Notably, the Alachua isolate carried a similar *Inc*Q plasmid (CP100656) that was 10,912bp, 1,428bp larger than the *Inc*Q plasmid (CP100659) identified from *S*. Senftenberg, with only a 3bp difference between the aligned regions of IncQ plasmids. Unlike the *Salmonella* Infantis ESI-CTX-M-65 cluster, The *S*. Senftenberg and *S*. Alachua strains do not carry a *gyr*A mutation (S2 Table).

Three ESS-CTX-M-65 of the early lineage without pESI plasmid were collected from North Carolina (Fig 5). Environmental isolates with the pESI plasmid were collected from either North Carolina (15/23), or Virginia (7/23), except 1 from South Carolina, which is located on a more recent lineage. Four isolates from human clinical samples were discovered at the later lineages and all carried pESI plasmid with *bla*CTX-M-65.

To investigate the possible location and time of horizontal transfer events of the pESI plasmids, a SNP analysis was conducted by mapping the raw reads from North Carolina and Virginia food animal isolates with pESI collected between 2019 and 2022 from ESI-CTX-M-65, ESS-CTX-M-65 and the single *Salmonella* Alachua isolate with pESI plasmid to a complete pESI plasmid pN16S024 sequence (CP052840.1). The phylogeny inferred from the SNP analysis shows that it is possible that the plasmid horizontally transferred to a *S*. Senftenberg strain around 2019 (Fig 6) from a turkey-derived *Salmonella* Infantis in a single event (Fig 6) in North Carolina. The Senftenberg isolate on the early branch likely harbored the *Inc*Q plasmid prior to the transfer event as shown in Fig 5.

The horizontal transfer event from a *S*. Senftenberg isolate to a *S*. Alachua isolate likely happened around 2020 because the isolate was collected in 2020 and all *Salmonella* Senftenberg were collected in 2020 or later (Fig 6). This *S*. Alachua isolate was also from comminuted turkey collected in North Carolina, USA, suggesting the event happened in North Carolina, USA as well.

The first human case infected with the *Salmonella* Senftenberg ESS-CTX-M-65 strain was reported in the PD Browser (PNUSAS276269) in June 2022. As of 1/23/2023, there were four clinical isolates carrying pESI plasmid with $bla_{\text{CTX-M-65}}$ gene (Fig 5).

## The comparison and evolution of the pESI plasmids from difference sources, serovars and years

The first reported pESI plasmid from Israel did not carry a $bla_{\text{CTX-M}}$ family gene and was identified in *Salmonella* Infantis isolates from humans [4]. Its length is smaller (285kb) when compared to later reported pESI plasmids (318 to 323kb) carried by S. Infantis from poultry [13]. The structure of the common regions (blue and orange) of pESI plasmids using pESI (CP047882) as the reference was shown in Fig 7. The backbone of the plasmids has three regions. The region with virulence gene clusters (highlighted with blue on the plasmid) is the most stable region, as there were only 20 positions with SNPs discovered between pESI (CP047882) as reference and five pESI plasmids isolated from difference source, year and serovars (S1 Table). In the clockwise direction, the sequence becomes more variable when it is further away from the *Inc*FIB replicon. Based on the pMLST scheme [27], the second region

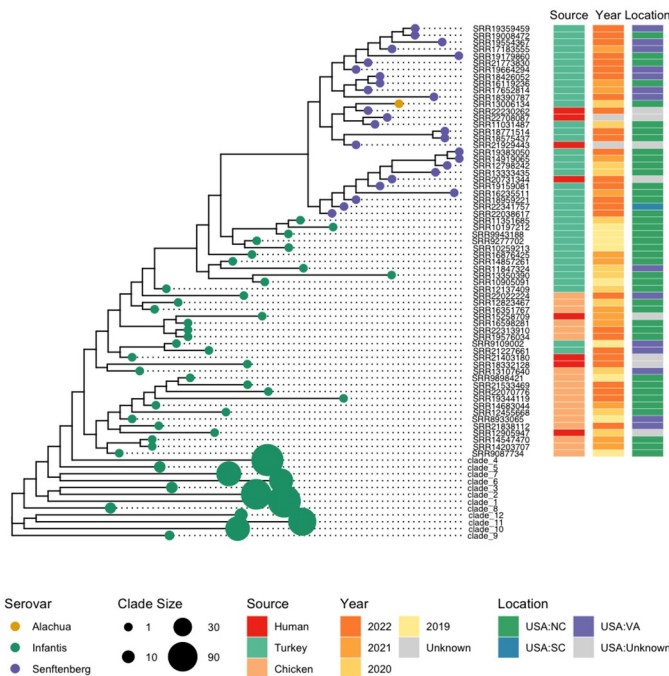

**Fig 6. The phylogenetic tree of pESI plasmid from food animal and human isolates.** The SNPs used to build the tree were from the CFSAN SNP pipeline by mapping the short reads to a complete pESI sequence (pN16S024, CP052840). Terminal branches that are not collapsed and which represent a single strain are identified by the SRR number. Branches that were collapsed are represented as larger circles, and labeled sequentially based on the size of the collapsed nodes.

labeled with orange originated from an *Inc*I1 plasmid. This region carries plasmid functional genes and there are more than 500 SNP locations. The third region, labeled with grey, composed of Integron *Int*I2 and its adjacent AMR genes and other genes with mostly unknown functions, is the most variable, with frequent insertions and deletions (Fig 7B). In fact, the variation of this region leads to most of the size difference among pESI plasmids. It is consistent with our observation that genes at region 2 (with *Int*I2) were most likely to be lost (Fig 4).

## Discussion

Plasmids contain many transposons and other repetitive elements that enable them to go through frequent recombination, which leads to gain or loss of AMR genes. However, all publications of pESI hosted by *Salmonella* Infantis from poultry reported a similar pattern of AMR genes [2, 5, 9, 12, 13, 15, 20]. In this study, we used the unique combination of AMR genes, including *bla*CTX, to search over a half million *Salmonella* in NCBI PD for the clusters of environmental strains potentially carrying pESI plasmids. In addition to the two well described *Salmonella* Infantis clusters ESI-CTX-M-65 [9, 12, 13, 19, 20] and ESI-CTX-M-1 [5, 9], we found two previously unreported *Salmonella* Infantis clusters carrying pESI ESI-CTX-M-14-1 and ESI-CTX-M-14-2 and two clusters of other serovars, although the isolates in some clusters had been described previously [11, 28]. Among them, *Salmonella* Alachua strain with pESI plasmid has never been reported before. Dos Santos et al. used a different strategy to search for any strains with pESI plasmid [23]. They first used virulence gene *ybt* to filter strains, and then used the core genome to identify the potential pESI. Besides the *Salmonella* Infantis strains, forty-three isolates of *Salmonella* Agona, Muenchen,

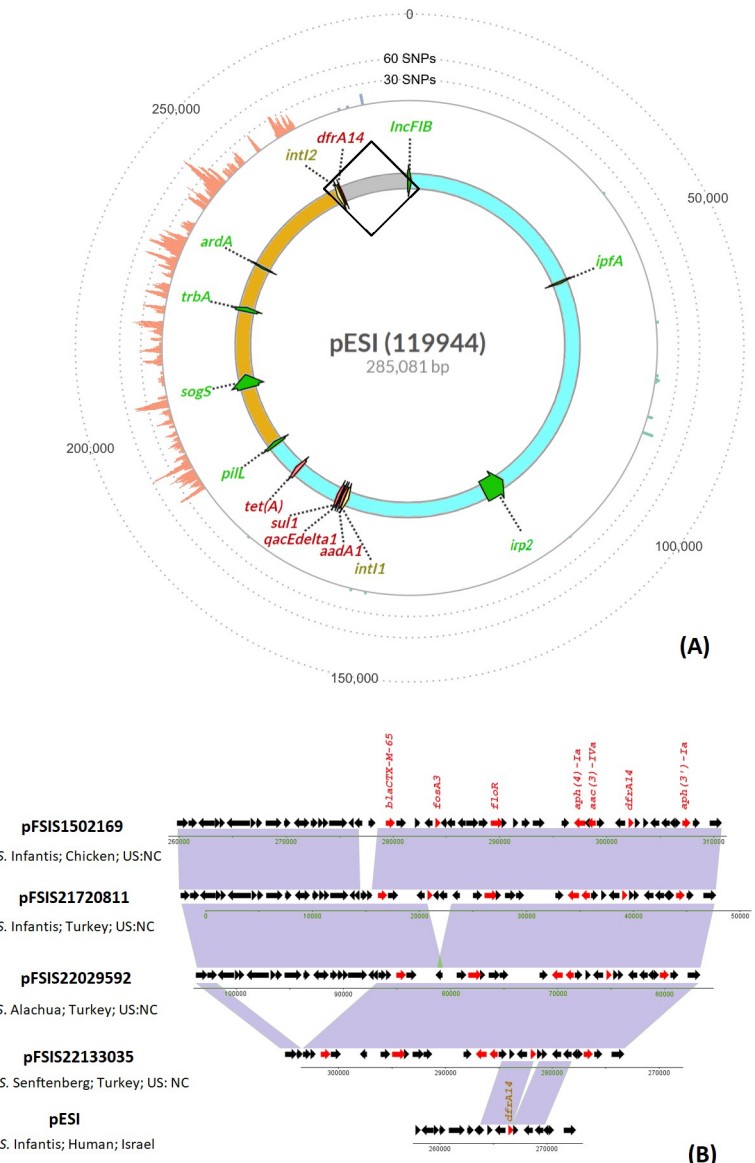

**Fig 7. The comparison of five pESI plasmid from different Salmonella serovars, host and years.** (A) The backbone of the pESIplasmid. The genes highlighted with green are seven markers used to screen for presence of pESI plasmids; The genes highlighted with red are antimicrobial AMR genes. The genes highlighted with yellow are two integrons. The outer track shows the number of SNPs when the four plasmids (pFSIS1502169, pFSIS21720811, pFSIS22029592, and pFSIS22133035) aligned to pESI plasmid, binned by 50 bp. On the pESI plasmid: the highlighted blue region carried virulence genes clusters encoding for fimbriae, yersiniabactin siderophore, toxin/anti-toxin and Integron IntI1 region; The highlighted orange region carried functional genes for plasmid conjugation, self-maintenance. (B) comparison of five pESI plasmids in the framed region of the pESI plasmid structure on (A). Source of the plasmids and accession numbers are listed in Table 2.

Senftenberg, Schwarzengrund were found to carry the pESI plasmid. Of note, the only environmental isolates in that study, three *Salmonella* Senftenberg isolates from turkey, were also identified in our study, using our strategy. It is possible that our selection strategy may have missed other clusters of isolates carrying a pESI plasmid without *bla*CTX-M gene. A strategy with higher sensitivity might need to be developed in future.

Our analysis aligns with previous findings that the ESI-CTX-M-65 clones with $bla_{CTX-M-65}$ circulated in the United States are likely to have originated from South America (Fig 1). This observation is consistent with a previous paper by Brown et al. [2], which reported that most early patients in the United States infected with $bla_{CTX-M-65}$-positive *Salmonella* Infantis had a history of traveling to South America.

Although only small proportion of the isolates in ESI-CTX-65 cluster were sequenced by long-reads, the resistance genes in Fig 4 are presumably located on the pESI plasmid since they are located on two conservative regions on the plasmid in almost all isolates with closed genomes with only one exception N18S2039, half of the plasmid with *bla*CTX-M-65 was integrated into the chromosome [13, 20].

We compared the average number of AMR genes carried by pESI from different regions. The increase in number of AMR genes in South and Central America around 2012 were caused by the insertion of additional AMR genes, including $bla_{CTX-M-65}$, in Region 2. These later isolates with more AMR genes, including $bla_{CTX-M-15}$, are presumably the ancestors of poultry isolates (Fig 2) from the United States. In the United States the average number of AMR genes in ESI-CTX-M-65 isolates/clones has decreased over time compared to their initial detection. The decline in average number of AMR genes among isolates from the United States is more prominent in strains from food animals than from humans (Fig 4B). The decline may be linked to decreases in the usage of antimicrobial drugs [29]. For example, the difference of reduction in *fos*A3 gene in isolates from food animal sources (0.17) and humans (0.10) (Fig 4B) may be due to the fact that fosfomycin is not approved for use in food animals in the United States (https://animaldrugsatfda.fda.gov/adafda/views/#/search). However, this type of reduction may be negatively impacted by frequent reintroduction of strains with higher number of AMR genes back into poultry.

Unlike Region 1 which is more stable, the structure of Region 2 makes it possible for the AMR cassette to lose some genes independently through recombination. Genes such as *fos*A3, *flo*R or $bla_{CTX-M-65}$ are surrounded by closely related repetitive genes (Fig 4. The prevalence of AMR genes at Region 1 remained above 80% through the years (Fig 7) and this may be due to the physical location of this region being closer to the *Inc*FIB(pN55391) replicon, unlike in Region 2, where AMR genes are physically located at the furthest end from the replicon, and may be prone to change. Alba et al. [24] compared five pESI plasmids which carried different blaCTX-M genes and isolated from 2013 to 2022, and concluded that although the sequences remained almost identical, their structures and resistance genes they carried can be different. Our study further illustrated the change of the pESI through the years.

The similarity of chromosomal and plasmid trees of Infantis ESI-CTX-M-65 strains in Fig 1A showed that the pESI plasmid spreads with clonal expansion. This observation is consistent to the previous reports [9, 19]. Furthermore, pESI in isolates from food animals in this study were found only in six clusters (number of clusters representing horizontal transfer events) out of which 95% of pESI were found in one ESI-CTX-M-65 cluster. It showed that the pESI likely spreads mainly from clonal expansion, with sporadic horizontal transfer.

The introduction of ESI-CTX-M-65 into food may have first occurred in chickens in the United States as the early isolates were mostly found in chicken. Isolates from turkey did not appear until 2016 (Fig 2). There were a few horizontal transfer events happening in the turkey processing environment in North Carolina, supported by plasmid phylogeny (Figs 5 and 6). Further epidemiological evidence is needed to support this theory. The ESS-CTX-M-65 strains appear to have become established in turkey since routine sampling detected this strain over multiple years. By comparison, another horizontal transfer event of pESI, occurred likely from Senftenberg to Alachua (Fig 5), however this did not lead to an immediate spread of pESI-containing Alachua, as from 2020 to date only one such strain has been reported. The exact reason

for this lack of spread of Alachua in turkeys despite the carriage of pESI will need further work.

In addition to pESI plasmid, *Salmonella* Alachua and a portion of the *Salmonella* Senftenberg carry an *Inc*Q plasmid, which carries up to four additional AMR genes (*sul*2, *aph*(3")-Ib, *aph*(6)-Id and *tet*(A)) (S1 Fig). It may be useful to investigate if the *Inc*Q plasmid facilitates with the transfer of the pESI plasmid.

During this study time period, some lineages shed all 10 AMR genes but still carried the pESI plasmid [13]. This indicates that the plasmid not only serves as an antimicrobial resistance gene reservoir, but it may also provide other competitive and fitness advantages to the host bacterium via its virulence genes encoding for fimbrial clusters and yesiniabactin siderophore. It is also interesting that 2% of the pESI plasmid showed high homology to the host chromosome (S2). These regions include genes coding for amino acid permease, arginine antiporter (*adi*C) and transposases (S2). Interestingly, these highly homologous regions are located in the region (highlight with blue in Fig 7A) originating from *Inc*FIB(pN55391plasmid. It may serve as the mechanism to ensure vertical transmission (i.e., each daughter cell receives a copy of the plasmid during cell replication). The arginine *adi*C gene is known to impart acid resistance and the ability to survive under extremely acidic conditions [30] and could have similar fitness function in *Salmonella* as in *E*. coli to survive under extremely acidic conditions [30], such as human digestion track. It might have been important that the strain can survive such environment before it acquires the plasmid through horizontal transferring.

The evolution of the plasmid does not seem to be a linear process as shown in Fig 1A. Along with point mutations, plasmids gained, exchanged and lost genes through recombination with other plasmids and the bacterial chromosome. For example, the highlighted orange region of pESI (Fig 7) is highly homologous to an epidemic plasmid ST71 IncI1. This region in pFSIS150269 is identical to 94% of a ST71 IncI1 plasmid pC271 (S3). The ST71 plasmid was reported to be polyclonally spreading in commensal *Escherichia coli* in Bolivia around 2011 [31]. Interestingly, the CTX-M-15 was predominant in CTX-M-producing *E*. coli before CTX-M-65 spread in Bolivia and Peru [31, 32], coinciding with the evolution of ESI-CTX-M-65. It is likely that pESI plasmid exchanged its contents with ST71 IncI1 in commensal environment through its spreading.

In summary, this study shows that ESI clone that has flourished in North America likely originated from South America and was introduced into the United States through multiple events and in different locations, and at different times. So far, the spread of pESI is mainly through clonal expansion. However, pESI can occasionally transfer horizontally to other clusters or different serovars. The success of the transfer event likely depends on what fitness advantage is conferred to the host strains from the newly acquired plasmid. Finally, individual AMR genes carried on the plasmid can be lost during clonal expansion and are mediated by recombination events in unstable areas of the plasmid in response to direct and/or indirect triggers.

The spread of the MDR pESI plasmid with ESBL gene $bla_{CTX-M}$ in the food production system posts a unique challenge to public health. Our study of pESI expansion and evolution will help public health scientists to understand, monitor and contain its spread through targeted mitigation strategies.

## Materials and methods

### The search for the carriages of pESI plasmid in *Salmonella*

The combined term of "AMR_genotypes:*dfr*A14 AND AMR_genotypes:*bla*CTX* AND AMR_genotypes:sul1 AND taxgroup_name:"*Salmonella* enterica" AND epi_type:

environmental*" was searched in 25222 clusters including 510,097 *Salmonella* isolates submitted to projects under NCBI pathogen detection browser (https://www.ncbi.nlm.nih.gov/pathogens/) before Jan 23th, 2023 for potential *Salmonella* clusters which carried pESI plasmid. Please note: Not all isolates within the cluster harbored the above genes.

## The confirmation of pESI plasmid

For clusters of interest, assemblies were downloaded. To confirm the presence of the pESI plasmid, the replicon sequence *Inc*FIB (pN55391) (CP016411), four evenly distributed markers on the *Inc*I region *ard*A, *pil*L, *Sog*S, *Trb*A, and the virulence genes *ipf* and *ipr*2 from pESI plasmid [5, 15] were queried against assemblies using BLAST 2.12 [33] The other markers used by others [5, 19] were either included in the search term, or close to the seven markers used in this study. The sequences of markers were extracted from a pESI plasmid pN16S024 (CP052840.1) [13] based on the primer sequences published by Franco et al [5], except for the replicon sequence, which was extracted directly from pN16S024 since the primer sequences were not included in the paper (S1 Table). The criteria for presence of a marker are greater than 85% identity and longer than 50% of coverage. The criterion for presence of the pESI plasmid is the presence of four markers or more.

## Data mining

We used *Salmonella* genomes deposited to BioProject 292661 (Contributed by Food and Drug Administration (FDA) National Antimicrobial Resistance Monitoring System (NARMS)), BioProjects 242847, 292666 (Contributed by the U.S. Department of Agriculture (USDA) NARMS program), and BioProject 230403 (Contributed by Centers for Disease Control and Prevention (CDC) NARMS program) to calculate the number of *Salmonella* isolates collected from retail meat, food animals at slaughter and human clinical sampling in the United States, respectively. All isolates from South and Central American countries were used to calculate the average number of AMR genes in South and Central America (Fig 3), regardless of the source.

All metadata, including geographic location of the sample, sample collection date, and AMR gene content were retrieved from the PD Browser. The resistance genes labeled as "MISTRANSLATION" were not counted. Duplicated resistance genes were individually checked and only one copy was counted as being present once because of the limitation of de-novo short read assemblies. The "human isolates" in this study refer to isolates collected from human clinical samples, and "environmental isolates" refer to isolates collected from animals, retail meat, water, and other environments.

## Phylogenetic analysis

In this study, only isolates from BioProjects 242847, 292666 and 230403 collected under the NARMS program were used to represent strains from the United States for phylogenetic analysis (Figs 1, 5 and 6). BioProject 230403 represents human clinical isolates, and BioProjects 242847 and 292666 represent isolates from food animals in the United States. Isolates from other countries were selected to represent different locations, sources, and years (Fig 1).

Sequencing reads were downloaded from the NCBI database using sratoolkit 3.0 (https://github.com/ncbi/sra-tools/wiki). CFSAN SNP pipeline (https://peerj.com/articles/cs-20/) with default parameter settings was used to construct the SNP matrix. The chromosome sequence of N16S024 (CP052839.1) was used as reference for mapping to find SNPs to construct chromosomal phylogeny of isolates from four different clusters with pESI plasmids (Fig 1). The plasmid sequence of N16S024 (CP052840.1) was used to find SNPs to construct plasmid

phylogeny of pESI (Figs 1 and 6). The chromosomal sequence of N18S0991(CP082574.1) was used as reference for phylogeny of ESS-CTX-M-65 (Fig 5). SNPs were called by mapping raw reads to the reference sequences using CFSAN SNP pipeline with default parameters. The phylogenetic tree was constructed using FastTree v.2.1 [34] with default parameters.

### Data visualization

The plot and bar figures (Figs 2–4) were constructed with ggplot2 [35] and ggpubr (https:// cran.r-project.org/web/packages/ggpubr/index.html) R packages; the tree was viewed and labeled by R package ggtree [36].

### DNA isolation short read and long read sequencing

Short-read WGS was performed at NARMS laboratories following GenomeTrakr (https:// www.protocols.io/workspaces/genometrakr1) and PulseNet protocols (https://www.cdc.gov/ pulsenet/pdf/PNL34-PN-Nextera-XT-Library-Prep-508.pdf). Sequencing was performed on 2886 clinical isolates under Bioproject 230403; 5827 food animal isolates under Bioprojects 242847 and 29266; and 925 isolates under Bioproject 292661. The procedure used at CVM was described previously [37]. The SRAs were submitted to NCBI pathogen detection database and were subsequently downloaded subsequentially for analysis, along with other SRAs deposit in PD database by other institutes. Three *Salmonella* isolates representing three separate serovars with pESI recovered from turkey from state of North Carolina were selected for long-read sequencing, including FSIS21720811 (*Salmonella* Infantis), FSIS22029592 (*Salmonella* Alachua) and FSIS22133035 (*Salmonella* Senftenberg). DNA extraction was done using DNeasy blood and tissue kits (Qiagen, Germantown, MD). The DNA was then quantified by Qubit fluorometer with dsDNA HS Assay kit (ThermoFisher Scientific, CA) per the manufacturer's instructions. The long-read sequencing was conducted as previously described [13, 38] using the 10-kb SMRTLink template preparation protocol. The library was sequenced on PacBio Sequel (Pacific BioSciences, CA). The long-read sequences were then assembled to complete genomes with Microbial Assembly pipeline embedded in SMRTLink 10.0. The complete genomes were annotated by NCBI Prokaryotic Genome Annotation Pipeline [39]. The details of these isolates and accession numbers are listed in Table 2.

### Supporting information

**S1 Table. Detailed information for all isolates in cluster ESI-CTX-M-65 (PDS000089910.296).** Tab "ESI-CTX-M-65_ PDS000089910.296" includes all isolates of the cluster and the metadata downloaded from NCBI Pathogen Broswer Database; Tab "pESI markers" includes the reference sequences of seven markers used to detect pESI; Tab "snplist Fig 7" shows the snp list generated by CFSAN snp pipeline, with pESI (CP047882.1) as reference and the column C is used to generate the snp map on Fig 7A.
(XLSX)

**S2 Table. Detailed information for all isolates in cluster ESI-CTX-M-1 (PDS000032463.103), ESI-CTX-M-14-1 (PDS000028342.9), ESI-CTX-M-14-2 (PDS000113177.3), and PDS000050501.** The metadata were downloaded from NCBI Pathogen Browser Database.
(XLSX)

**S1 Fig. Comparison between two IncQ plasmids pFSIS22029592-2 (CP100656.1) and pFSIS22133035-2 (CP100659.1).**
(TIF)

**Table 2. The isolates information in Fig 7.**

| Isolate ID | Serovar | Country | State | Year | Sample Type | Source of Isolation | Collected by | pESI plasmid | Length (bp) | Accession |
|---|---|---|---|---|---|---|---|---|---|---|
| 119944 | *Salmonella* Infantis | Israel | Israel | 2008 | Clinical | Human | Infectious Diseases Research Laboratory | pESI | 285,081 | NZ_CP047882 |
| FSIS1502169 | *Salmonella* Infantis | United States | North Carolina | 2015 | Environmental | Chicken | USDA-FSIS | pFSIS1502169 | 323,162 | CP016407 |
| FSIS21720811* | *Salmonella* Infantis | United States | North Carolina | 2017 | Environmental | Comminuted Turkey | USDA-FSIS | pFSIS21720811 | 322,470 | AANAOP020000003 |
| FSIS22029592* | *Salmonella* Alachua | United States | North Carolina | 2020 | Environmental | Comminuted Turkey | USDA-FSIS | pFSIS22029592 | 318,597 | CP100655 |
| FSIS22133035* | *Salmonella* Senftenberg | United States | North Carolina | 2021 | Environmental | Comminuted Turkey | USDA-FSIS | pFSIS22133035 | 303553 | CP100658 |

\* The complete genomes of the isolates were generated in this study.

**S2 Fig. Alignment between plasmid pN16S024 (CP052840.1) and chromosome N16S024 (CP052839.1).** The Alignment is generated by Mauve (https://darlinglab.org/mauve/mauve.html). The color blocks show the regions with homology over 80%.
(TIF)

**S3 Fig. Alignment between plasmid pC271 (NZ_LN735561.1) and pN16S024 (CP052839.1).** The Alignment is generated by Mauve (https://darlinglab.org/mauve/mauve.html). The color blocks show the regions with homology over 80%.
(TIF)

## Acknowledgments

We thank Amy Merrill and Ryan McDonald for data analysis. We thank Glenn Tillman and FSIS Laboratory staff for contributing FSIS *Salmonella* to this study. We also thank the Pulse-Net participating laboratories for isolating and sequencing the Salmonella isolates, uploading the sequences to the PulseNet Salmonella National Database, and submitting the raw sequence data to the NCBI public databases. We wish to thank the National Antimicrobial Resistance Monitoring System at the Centers for Disease Control and Prevention for their role in public health surveillance for human infections harboring pESI plasmids.

**Author disclaimer:** The views expressed in this manuscript are those of the authors and do not necessarily reflect the official policy of the Department of Health and Human Services, the U.S. Food and Drug Administration, or the U.S. Government. Reference to any commercial materials, equipment, or process does not in any way constitute approval, endorsement, or recommendation by the Food and Drug Administration.

## Author Contributions

**Conceptualization:** Cong Li, Errol A. Strain.

**Formal analysis:** Cong Li, Errol A. Strain.

**Funding acquisition:** Patrick F. McDermott, Errol A. Strain.

**Investigation:** Cong Li, Uday Dessai, Errol A. Strain.

**Methodology:** Cong Li.

**Project administration:** Heather Tate, Gamola Z. Fortenberry, Uday Dessai, Patrick F. McDermott, Errol A. Strain.

**Resources:** Gamola Z. Fortenberry, Uday Dessai, Patrick F. McDermott, Errol A. Strain.

**Software:** Chih-Hao Hsu, Errol A. Strain.

**Supervision:** Errol A. Strain.

**Visualization:** Xinyang Huang, Chih-Hao Hsu.

**Writing – original draft:** Cong Li, Heather Tate, Lucas B. Harrison, Shaohua Zhao, Patrick F. McDermott, Errol A. Strain.

**Writing – review & editing:** Cong Li, Heather Tate, Lucas B. Harrison, Errol A. Strain.

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
