## [Decision Letter · Decision Letter 0]

6 Sep 2023

PONE-D-23-23845The spread of pESI plasmids among Salmonella Serovars - Infantis, Senftenberg, and Alachua isolated from food animal sources in the United StatesPLOS ONE

Dear Dr. Li

Thank you for submitting your manuscript to PLOS ONE. After careful consideration, we feel that it has merit but does not fully meet PLOS ONE’s publication criteria as it currently stands. Therefore, we invite you to submit a revised version of the manuscript that addresses the points raised during the review process.

Please submit your revised manuscript by Sep.21, 2023 .  If you will need more time than this to complete your revisions, please reply to this message or contact the journal office at plosone@plos.org. Please include the following items when submitting your revised manuscript:A rebuttal letter that responds to each point raised by the academic editor and reviewer(s). You should upload this letter as a separate file labeled 'Response to Reviewers'.A marked-up copy of your manuscript that highlights changes made to the original version. You should upload this as a separate file labeled 'Revised Manuscript with Track Changes'.An unmarked version of your revised paper without tracked changes. You should upload this as a separate file labeled 'Manuscript'.

We look forward to receiving your revised manuscript.

Kind regards,

Maria Pia Franciosini, Ph.DVM

Academic Editor

PLOS ONE

Journal Requirements:

3. Ethics statement does not appear in the manuscript file:

Please include your full ethics statement in the ‘Methods’ section of your manuscript file. In your statement, please include the full name of the IRB or ethics committee who approved or waived your study, as well as whether or not you obtained informed written or verbal consent. If consent was waived for your study, please include this information in your statement as well.

Additional Editor Comments:

Although is an original article, major changes are required .Please try to meet the expert remarks , where possible, including a point by point reply to their comments . Look also carefully since some minor flaws are scattered throughout the text. Finally try to follow the guidelines required for article submission

Reviewers' comments:

Reviewer's Responses to Questions

**Comments to the Author**

1. Is the manuscript technically sound, and do the data support the conclusions?

Reviewer #1: No

Reviewer #2: Partly

2. Has the statistical analysis been performed appropriately and rigorously? 

Reviewer #1: I Don't Know

Reviewer #2: No

3. Have the authors made all data underlying the findings in their manuscript fully available?

Reviewer #1: No

Reviewer #2: No

4. Is the manuscript presented in an intelligible fashion and written in standard English?

Reviewer #1: Yes

Reviewer #2: Yes

5. Review Comments to the Author

Reviewer #1: Consider suggestions to clarify this manuscript.

Line 129 authors state the Table 1 shows that each cluster carried a unique blaCTX-M allele.

But blaCTX-M-65, blaCTX-M-15, blaCTX-M-55, blaCTX-M-14, are each found in multiple clusters directly contradicting their observation and statement.

Line 374 and earlier, investigators state Fig 1A showed that the pESI plasmid spreads with clonal expansion but there are many crossovers and disagreements between the two trees that are typically interpreted as horizontal transfer and or rearrangements. So how do the authors explain this contradiction in their interpretation? Line 407, The authors seem to contradict themselves with this statement, The evolution of the plasmid does not seem to be a linear process as shown in Fig 1A. which I agree with.

Figure 1 is unreadable the authors should present a figure where the reviewer can read the isolate name on each terminal branch.

Figures 5, 6 the investigators should label all terminal branches with unique isolate names.

Data reproducibility, the authors should list all unique NCBI SRA genome ids for each isolate presented in their figures. These ids should match terminal identifications listed on their figures.

Reviewer #2: General Comments

S. Infantis harbouring pESI is a matter of great concern because this megaplasmid has been found to carry several (from 3 to 6) antimicrobial resistance genes, resistance genes to heavy metals and disinfectants and also other virulence factors enhancing the fitness of S. Infantis.

In this manuscript entitled “The spread of pESI…”, the authors aimed to i) understand the evolution of pESI harbouring blaCTX-M-65 by comparing using SNPs analysis all the S. Infantis sequences likely carrying pESI plasmids that were available at NBCI’s PD database at the time of the study; ii) investigate the genetic changes of the American variant of pESI-like plasmid in ESI clones . Moreover and importantly, they state they have detected other pESI-like-positive serovars beside S. Infantis, pointing out that pESI(like) spread could be associated to horizontal transfer events other than clonal expansion, as previously demonstrated.

However, some aspects about the methods used and the results obtained need to be further clarified and the discussions ensued around them, whenever necessary, should be moderated. In some cases, similar findings described in this manuscript have been already reported in recent published papers, which neither have been cited in the main text nor are present in the reference list provided by the Authors. These papers should be properly cited, as the novelties and main findings of this study should be discussed in comparison with what has been previously published (see specific comments).

Specific Comments:

- The Authors should acknowledge a possible bias in the selection of data/genomes. From Materials & Methods, only MDR S. Infantis with blaCTX-M-65 were investigated. Otherwise, the reader could understand that 99 % of the S. Infantis in the Americas were pESI-positive, which would be worrisome.

- Along the manuscript, authors used terminology like “evolution” or “recent common ancestor”. It is necessary to know how the authors include the isolation time by using the FastTree phylogeny tool. Then in any case this represents a bias when dealing with “evolution” and “ancestry”, as the Authors have included in the study only isolates with plasmids carrying resistance to third-fourth generation cephalosporins, which is a feature that only a subset of isolates around the world do possess.

- In any case, from the data presented, analysed and discussed, the title of the manuscript should be rephrased, in order to avoid overstatements. Here is a proposal: “The spread of pESI-mediated extended-spectrum cephalosporin resistance in Salmonella Serovars - Infantis, Senftenberg, and Alachua isolated from food animal sources in the United States.” Most importantly, the title should reflect certainty on the key feature “pESI-mediated”, since (see also below) in many cases the Authors do not bring data demonstrating that all ESC resistance genes (CTX-M genes) in all serovars/isolates considered are harboured by such pESI plasmids. Otherwise, the title should be “The spread of pESI in ESBL-producing Salmonella Serovars - Infantis, Senftenberg, and Alachua isolated from food animal sources in the United States”.

- Lines 75-76: “Our previous work and other studies have shown that pESI are transmitted mainly through clonal expansion (5, 9, 13)”. References 5 and 9 are not correctly cited here. Authors in Alba et al., 2020 reported that the pESI-like success and spread could be the result of a mixed effect of clonal expansion and horizontal transfer: “…once the megaplasmid is acquired by S. Infantis, it would likely quickly spread in the local S. Infantis population because of the presence of the specific plasmid-borne genes, which enhance the colonization, virulence and fitness behaviour of the strain…These genetic elements play a central role in bacterial adaptability in response to stress conditions and in the maintenance of plasmids or genomic islands [43], also promoting the ecological success of certain clones, such as the pESI- like- positive- ESBL- producing S. Infantis clone..”. Please rephrase your sentence, accordingly.

- Line 86: the presence of pESI-like in S. Senftenberg from US samples has been previously reported in a large-scale genomic analysis conducted on S. Infantis sequences retrieved from NCBI by dos Santos et al., 2022 (https://www.sciencedirect.com/science/article/pii/S0740002022001368?via%3Dihub). In particular, they have found pESI markers in four different serovars: S. Muenchen, S. Schwarzengrund, S. Agona and S. Senftenberg. This paper should be properly cited and included in the discussion. Moreover, to the best of the Reviewer's knowledge the present study represents the first documented report of the presence of pESI in S. Alachua. In this case, the novelty may be highlighted in the main text.

- Line 168-172: In this paragraph and in the figure 1, the location of blaCTX-M-15 in the genome has not been clarified. The authors should check and report the location of blaCTX-M-15 gene. To the best of the reviewer’s knowledge, it has not been described any pESI harbouring bla-CTX-M-15, yet.

- In general, and most importantly (see above) they should clarify and report the location of the blaCTX-M genes in all the studied pESI-like plasmids (in some cases it has been demonstrated in previous studies and for specific serovars, but not in all pESI (like) sequences).

- Figure 4. Additional information on how the prevalence of AMR genes for each genetic region has been calculated is missing. For example, how the normalization was done and how the authors decide from short-reads sequencing data which genes were in those regions…

- Line 301: How have the authors calculated the timing? How about the mutation rate? This should be clearly declared in the M&M section.

- Line 323: the rearrangements and structural variation in the different pESI-like plasmid variants (blaCTX-M-65, blaCTX-M-1-positive and blaCTX-M negative ones) were already demonstrated in Alba et al., 2023 (https://academic.oup.com/femsle/article/doi/10.1093/femsle/fnad014/7049104?login=false) . The authors should include this paper in the references. Do their findings confirm what has been reported in this recent publication?

- Line 345: Please cite Alba et al., 2020 (9) together with Franco et al., 2015 (5) for the ESI-CTX-M-1 cluster (in this case the Authors of the submitted manuscript refer to the bacterial host, not to the plasmid, if I am correct….).

- Line 346: Although I agree with the Authors that S. Infantis clusters harbouring blaCTX-M-14 have not been previously reported, pESI harbouring blaCTX-M-14 has been previously described in Bogomazova et al., 2020 and Egorova et al., 2023 (https://www.mdpi.com/2076-2607/11/2/347). This latter paper should be properly cited and included in the reference list.

- Lines 347-349 “Our analysis…nearby countries”. This sentence concerning the “likely” origin of the ESI CTX-M-65 clone is quite speculative, also considering the available literature and should be deleted. Otherwise, the Authors should bring adequate data.

- Line 381: this was stated by McMillan et al. 2022: the authors should cite the paper here.

- Line 399 – 406 : To validate this hypothesis, authors should verify if these genes were present or absent in the chromosome of the isolates harbouring pESI with these genes. If the gene is already present in the chromosome, its presence within the plasmid is not so important for the fitness of S. Infantis. Moreover, in isolates harbouring blaCTX-M-1, this gene is close to blaCTX-M-1 but distant from the replicon.

- Line 477 and following: “SNPs were called…”. Could the authors clarify this point, explaining if they have used different methods: one for all the S. Infantis and one for the ESS-CTX-M-65 phylogeny or if bowtie2 is part of the CFSAN SNP pipeline?

- Line 484 The authors should add some information about the statistics (or math in general) used to get the numbers to build the figures

- Line 486: Could the authors declare the number of isolates WGSed?

- Lines 489-492 “Three Salmonella… Senftenberg”: the authors declared they have sequenced three Salmonella isolates (including S. Senftemberg and S. Alachua) by long read-sequencing. How did they demonstrate the presence of pESI plasmids (other than the presence of pESI markers) in these other two Salmonella serovars using the long-read sequencing approach? How did they locate blaCTX-M-65 and other AMR genes on these pESI plasmids? These aspects should be clearly reported in the M&M and Results section.

- Line 498: Authors should declare the accession number of the new sequences (assembly or raw reads) obtained in this study.

- Supplementary file S2: C140670-sal, 17031040-sal, 16023650-sal isolates should be removed from spreadsheet “ESI-CTX-M-1_PDS000032463.92”as they do not harbour blaCTX-M-1 (see Supplementary table of Alba et al., 2020). The Authors should also check if the isolate UZH-SAL-119-15 from Switzerland harbour blaCTX-M-1. I suggest to re-check the AMR gene content of all the sequences reported in the Supplementary file S2.

6. PLOS authors have the option to publish the peer review history of their article (what does this mean?). If published, this will include your full peer review and any attached files.

Reviewer #1: No

Reviewer #2: No

---

## [Author Response · Author response to Decision Letter 0]

22 Sep 2023

To Reviewer #1:

Thanks for your thorough review! The questions really help us to explain the main ideas better. The revised figures are clearer.

To Reviewer #2:

Thanks for your thorough review! Your questions and recommendations for change of the title, additional publications etc. have certainly made this paper much better! We really appreciate it.

---

## [Decision Letter · Decision Letter 1]

25 Oct 2023

PONE-D-23-23845R1The spread of pESI-mediated extended-spectrum cephalosporin resistance in Salmonella serovars - Infantis, Senftenberg, and Alachua isolated from food animal sources in the United StatesPLOS ONE

Dear Dr. Li,

Thank you for submitting your manuscript to PLOS ONE. After careful consideration, we feel that it has merit but does not fully meet PLOS ONE’s publication criteria as it currently stands. Therefore, we invite you to submit a revised version of the manuscript that addresses the points raised during the review process.

We look forward to receiving your revised manuscript.

Kind regards,

Maria Pia Franciosini, Ph.DVM

Academic Editor

PLOS ONE

**Additional Editor Comments:**

I appreciated your efforts in meeting   remarks of reviewers  due to which your manuscript is greatly improved but take into consideration that further major changes are still  required by a reviewer.  When you cannot  address reviewer's remark, please  justify your point of view in the cover letter, as specified below.

Reviewers' comments:

Reviewer's Responses to Questions

**Comments to the Author**

1. If the authors have adequately addressed your comments raised in a previous round of review and you feel that this manuscript is now acceptable for publication, you may indicate that here to bypass the “Comments to the Author” section, enter your conflict of interest statement in the “Confidential to Editor” section, and submit your "Accept" recommendation.

Reviewer #1: All comments have been addressed

Reviewer #2: (No Response)

2. Is the manuscript technically sound, and do the data support the conclusions?

Reviewer #1: Yes

Reviewer #2: Partly

3. Has the statistical analysis been performed appropriately and rigorously? 

Reviewer #1: Yes

Reviewer #2: No

4. Have the authors made all data underlying the findings in their manuscript fully available?

Reviewer #1: Yes

Reviewer #2: Yes

5. Is the manuscript presented in an intelligible fashion and written in standard English?

Reviewer #1: Yes

Reviewer #2: Yes

6. Review Comments to the Author

Reviewer #1: no additional edits recommended by this reviewer. the authors have address the edits suggested to be closely examined. no additional edits recommended by this reviewer. the authors have address the edits suggested to be closely examined.

Reviewer #2: The manuscript has been modified and it seems that most of the reviewer’s comments have been evaluated by the authors and included in the new version of the manuscript. However, the three big “issues” that needed to be addressed, have not been resolved yet (or only slightly modified in the present form).

-The bias of the isolate/genome selection has now been declared in the discussion part, but it has not been considered when formulating the conclusions. For example, in line 169-171 (tracked changes version), the Authors have stated: “If the evolution of the plasmids was not affected by the host, it would not form clades that are similar in structure…”. The authors should consider whether the cluster topology would change if other S. infantis isolates (CTX-M-neg or pESI neg) were included. This is important, as this phenomenon (agreement of chromosomal SNP tree and plasmid SNP tree) has not been observed in the previous literature on the topic (Alba et al., 2020).

-The statement that ESI-CTX-M-65 derived from ESI-CTX-M15 is an overstatement because there is no evidence of it in the Authors’ experiments. Maybe, from the fig 1 (without knowing the number of SNPs differences or the length of the branch) you can say/infer that the chromosomal content is very similar, but not the plasmid tree. In their own figure, the Authors can observe how huge the branch length on the plasmid tree is. If they write on the manuscript that ESI CTX-M-65 derived from ESI CTX-M-15, the general educated reader could think that the plasmid had evolved from a CTX-M-15-positive pESI, but there is no evidence of it from this study. From their data, I have interpreted that the clone (chromosome) is similar (one would need to know the SNPs difference) but not the plasmid sequence. Maybe the CTX-M-15 had pESI without blactx-m-15 and blactx-m-15 could either be located in other plasmids or this isolate could harbor only the IncI half part of the pESI plasmid.

And, in any case, and most importantly, if S. Infantis did not harbour pESI, it should not be named ESI.

-The Authors answered that the reconstruction of timing of pESI introduction and evolution was based on S. infantis isolation time from metadata. This is not enough information to talk about “evolution”. The isolation time let one speculate on possible scenario/s, however it does not provide a “probabilistic evolutionary model”, that need to be calculated to predict when pESI was introduced and then evolved in the S. Infantis population described in this study. The Authors should not refer and use terms like “evolution” throughout the text, unless they provide evidence on how it was calculated, in particular when they refer to pESI plasmid. This also represents a limitation of this study that should be clearly underlined in the discussion section.

Response to Minor comments:

-The isolate C140670-sal to which I refer was included in the study Alba et al., 2020 (PRJEB23728) where its sequence was originally included.

-Isolates 17031040-sal, 16023650-sal, UZH-SAL-119-15 (CTX-M negative) were included in a spreadsheet named “ESI-CTX-M-1_PDS000032463.92”. Could the authors clarify what kind of sequences are included in the different spreadsheets of Supplementary File S2? What is the common feature among these groups of sequences? In case, I suggest to rename the different spreadsheets, accordingly.

7. PLOS authors have the option to publish the peer review history of their article (what does this mean?). If published, this will include your full peer review and any attached files.

Reviewer #1: No

Reviewer #2: No

---

## [Author Response · Author response to Decision Letter 1]

3 Nov 2023

Dear Reviewer, 

Thanks for your careful review and the detailed comments. I do feel it makes the paper a lot more readable for general scientists in the food safety and microbiology field. We really appreciate your hard work. 

Best,

Cong

---

## [Decision Letter · Decision Letter 2]

30 Jan 2024

PONE-D-23-23845R2The spread of pESI-mediated extended-spectrum cephalosporin resistance in Salmonella serovars - Infantis, Senftenberg, and Alachua isolated from food animal sources in the United StatesPLOS ONE

Dear Dr. Li,

Thank you for submitting your manuscript to PLOS ONE. After careful consideration, we feel that it has merit but does not fully meet PLOS ONE’s publication criteria as it currently stands. Therefore, we invite you to submit a revised version of the manuscript that addresses the points raised during the review process.

We look forward to receiving your revised manuscript.

Kind regards,

Maria Pia Franciosini, Ph.DVM

Academic Editor

PLOS ONE

Journal Requirements:

Additional Editor Comments:

Dear Authors,

You have definitely improved the manuscript, addressing most of comments both in the revised copy and in point-by-point response letters for the reviewers . However some minor changes are still required for pubblication. Please consider the following reviewer's comments

The authors have compiled a well-written manuscript that adds to the narrative of the spread of pESI-mediated extended-spectrum cephalosporin resistance in Salmonella serovars - Infantis, Senftenberg, and Alachua in the United States. However, from the narrative, it is not totally clear to me at the beginning if it was an actual study where isolates were analyzed or if sequences were retrieved from NCBI for the analysis. It is a bit confusing

At a certain point, it reads as if data were retrieved from the database and at the method sections it reads as if you conducted the actual test. Please clarify line 520. Though some isolates were received from NARMs it is not clear at the beginning of the manuscript.

Line 72-73 “Additionally, some Salmonella Infantis strains with pESI-like plasmid have been found carrying a colistin-resistance gene (mcr-1) (17, albeit not on the pESI-like plasmid, further limiting treatment options”. Could you please clarify this statement as the understanding is not clear?

Line 92 in this study is this a study or review

Line 128-134 Only 13 out of 5765 (0.2%) blaCTX-M alleles are blaCTX-M-15, 5687 (98.6%) are blaCTX-M-65, the remaining 65 are unnamed blaCTX-M alleles (S1Table). It seems the numbers do not add up. Can you explain where these numbers are coming from?

Line 192 I am not sure where 8935 is coming from it is not on the table “143 of the 8,935 isolates in this cluster with”

Line207 can you specifics how many strains from NARM was used

Line 446 E. coli should be italicized

Line 503. “Isolates from other countries were selected randomly based on location, source, and year” This is not clear to me. kindly indicate where are the isolates coming from and why that selections.

Reviewers' comments:

Reviewer's Responses to Questions

**Comments to the Author**

1. If the authors have adequately addressed your comments raised in a previous round of review and you feel that this manuscript is now acceptable for publication, you may indicate that here to bypass the “Comments to the Author” section, enter your conflict of interest statement in the “Confidential to Editor” section, and submit your "Accept" recommendation.

Reviewer #2: (No Response)

Reviewer #3: All comments have been addressed

Reviewer #4: All comments have been addressed

2. Is the manuscript technically sound, and do the data support the conclusions?

Reviewer #2: Partly

Reviewer #3: Yes

Reviewer #4: Yes

3. Has the statistical analysis been performed appropriately and rigorously? 

Reviewer #2: No

Reviewer #3: Yes

Reviewer #4: Yes

4. Have the authors made all data underlying the findings in their manuscript fully available?

Reviewer #2: Yes

Reviewer #3: Yes

Reviewer #4: Yes

5. Is the manuscript presented in an intelligible fashion and written in standard English?

Reviewer #2: Yes

Reviewer #3: Yes

Reviewer #4: Yes

6. Review Comments to the Author

Reviewer #2: The Authors have not addressed major comments on the R2 version of the manuscript, so the study limits and overstatements i have observed are still present in the main text.

Reviewer #3: The study of pESI expansion and evolution would help public health scientists to understand, monitor and contain its spread through targeted mitigation strategies.

L94 and 98, PD need full spelling when first appear.

L99, NARMS need full spelling.

L118 six clusters carried pESI plasmid, why to choose this? Need proofs.

L123 2023 year?

L315 add USA

Reviewer #4: The authors have compiled a well-written manuscript that adds to the narrative of the spread of pESI-mediated extended-spectrum cephalosporin resistance in Salmonella serovars - Infantis, Senftenberg, and Alachua in the United States. However, from the narrative, it is not totally clear to me at the beginning if it was an actual study where isolates were analyzed or if sequences were retrieved from NCBI for the analysis. It is a bit confusing

At a certain point, it reads as if data were retrieved from the database and at the method sections it reads as if you conducted the actual test. Please clarify line 520. Though some isolates were received from NARMs it is not clear at the beginning of the manuscript.

Line 72-73 “Additionally, some Salmonella Infantis strains with pESI-like plasmid have been found carrying a colistin-resistance gene (mcr-1) (17, albeit not on the pESI-like plasmid, further limiting treatment options”. Could you please clarify this statement as the understanding is not clear?

Line 92 in this study is this a study or review

Line 128-134 Only 13 out of 5765 (0.2%) blaCTX-M alleles are blaCTX-M-15, 5687 (98.6%) are blaCTX-M-65, the remaining 65 are unnamed blaCTX-M alleles (S1Table). It seems the numbers do not add up. Can you explain where these numbers are coming from?

Line 192 I am not sure where 8935 is coming from it is not on the table “143 of the 8,935 isolates in this cluster with”

Line207 can you specifics how many strains from NARM was used

Line 446 E. coli should be italicized

Line 503. “Isolates from other countries were selected randomly based on location, source, and year” This is not clear to me. kindly indicate where are the isolates coming from and why that selections.

7. PLOS authors have the option to publish the peer review history of their article (what does this mean?). If published, this will include your full peer review and any attached files.

Reviewer #2: No

Reviewer #3: **Yes: **Qingli Dong

Reviewer #4: No

---

## [Author Response · Author response to Decision Letter 2]

6 Feb 2024

Thanks for all comments check and edits. We really appreciate it. It does help to improve the manuscript.

Best,

Cong

---

## [Editor Report · Decision Letter 3]

9 Feb 2024

The spread of pESI-mediated extended-spectrum cephalosporin resistance in Salmonella serovars - Infantis, Senftenberg, and Alachua isolated from food animal sources in the United States

PONE-D-23-23845R3

Dear Dr. Cong Li

We’re pleased to inform you that your manuscript has been judged scientifically suitable for publication and will be formally accepted for publication once it meets all outstanding technical requirements.

An invoice for payment will follow shortly after the formal acceptance. To ensure an efficient process, please log into Editorial Manager at http://www.editorialmanager.com/pone/, click the 'Update My Information' link at the top of the page, and double check that your user information is up-to-date. If you have any billing related questions, please contact our Author Billing department directly at authorbilling@plos.org.If your institution or institutions have a press office, please notify them about your upcoming paper to help maximize its impact. If they’ll be preparing press materials, please inform our press team as soon as possible -- no later than 48 hours after receiving the formal acceptance. Your manuscript will remain under strict press embargo until 2 pm Eastern Time on the date of publication. For more information, please contact onepress@plos.org.

Kind regards,

Maria Pia Franciosini, Ph.DVM

Academic Editor

PLOS ONE

---

## [Editor Report · Acceptance letter]

6 Mar 2024

PONE-D-23-23845R3 

PLOS ONE

Dear Dr. Li, 

I'm pleased to inform you that your manuscript has been deemed suitable for publication in PLOS ONE. Congratulations! Your manuscript is now being handed over to our production team.

Kind regards, 

on behalf of

Professor Maria Pia Franciosini 

Academic Editor

PLOS ONE